# Independent and Combined Association of Lifestyle Behaviours and Physical Fitness with Body Weight Status in Schoolchildren

**DOI:** 10.3390/nu14061208

**Published:** 2022-03-12

**Authors:** Rubén Aragón-Martín, María del Mar Gómez-Sánchez, José Manuel Martínez-Nieto, José Pedro Novalbos-Ruiz, Carmen Segundo-Iglesias, María José Santi-Cano, José Castro-Piñero, Carmen Lineros-González, Mariano Hernán-García, Mónica Schwarz-Rodríguez, David Jiménez-Pavón, Amelia Rodríguez-Martín

**Affiliations:** 1Department of Biomedicine, Biotechnology and Public Health, Faculty of Nursing and Physiotherapy, University of Cádiz, 11009 Cádiz, Spain; ruben.aragon@uca.es (R.A.-M.); mar.gomez@uca.es (M.d.M.G.-S.); josepedro.novalbos@uca.es (J.P.N.-R.); amelia.rodriguez@uca.es (A.R.-M.); 2Previene-Cádiz, European ITI Project PI-0007-2017, Andalusian Operational Program FEDER (European Regional Development Fund) 2014–2020, 11009 Cádiz, Spain; josemanuel.martinez@uca.es (J.M.M.-N.); csegundoiglesias@gmail.com (C.S.-I.); mariajose.santi@uca.es (M.J.S.-C.); jose.castro@uca.es (J.C.-P.); carmen.lineros.easp@juntadeandalucia.es (C.L.-G.); easp15@gmail.com (M.H.-G.); monica.schwarz@uca.es (M.S.-R.); 3Biomedical Research and Innovation Institute of Cádiz (INiBICA) Research Unit, Puerta del Mar University Hospital, University of Cádiz, 11009 Cádiz, Spain; 4MOVE-IT Research Group, Department of Physical Education, Faculty of Education Sciences, University of Cádiz, 11519 Puerto Real, Spain; 5Department of Nursing and Physiotherapy, Faculty of Nursing and Physiotherapy, University of Cádiz, 11009 Cádiz, Spain; 6Salus Infirmorum Nursing School, University of Cádiz, 11001 Cádiz, Spain; 7GALENO Research Group, Department of Physical Education, Faculty of Education Sciences, University of Cádiz, 11519 Puerto Real, Spain; 8Andalusian School of Public Health, 18080 Granada, Spain; 9Andalusian Council of Childhood, 18001 Granada, Spain; 10Nutrition and Bromatology Area, Faculty of Medicine, University of Cádiz, 11003 Cádiz, Spain; 11CIBER of Frailty and Healthy Aging (CIBERFES), 28029 Madrid, Spain

**Keywords:** childhood obesity, healthy lifestyle, physical fitness, body mass index, physical activity, sedentary behaviour, diet, sleep, screen time

## Abstract

(1) Background: Lifestyle behaviours and physical fitness play a critical role in the development of childhood obesity. It has been demonstrated in this study that self-reported physical fitness is representative of a healthy lifestyle and thus is associated with a lower incidence of overweight/obesity. The objective of this study was to analyse the independent and combined association of lifestyles (physical activity, screen time, diet and hours of sleep) and self-reported physical fitness with body weight in schoolchildren. (2) Methods: This study performed a descriptive and cross-sectional analysis. The study sample consisted of 864 schoolchildren between 8–9 years old from 26 schools of the province of Cádiz. To measure lifestyles and self-reported physical fitness, questionnaires were administered to both schoolchildren and families. To obtain the body weight status, the children were measured by body mass index (BMI). To verify the influence of lifestyles and self-reported physical fitness on the body weight status of schoolchildren, a combined score of lifestyles and self-reported physical fitness was calculated. (3) Results: Schoolchildren who followed healthier lifestyles and presented good physical fitness had a better body weight status (*p* < 0.001). Schoolchildren who had less healthful lifestyles and bad physical fitness had a 10.34 times higher risk of being overweight or obese (*p* = 0.004). (4) Conclusions: It has been shown that there is an independent and combined association between lifestyles and physical fitness on the body weight of the schoolchildren. We have suggested strategies to get children to adopt healthy lifestyles and good physical fitness to maintain a healthy body weight and prevent obesity.

## 1. Introduction

The World Health Organization (WHO) describes childhood obesity as one of the most important public health problems in developed countries [1]. However, it is not only epidemic in developed countries, such as western European countries, Australia, Canada and the USA, but also in developing countries, such as Mexico, Chile and China [2]. Mexico reported in 2016 a combined prevalence of obesity and overweight of 33.2% in children [3]. Around 23% of children and adolescents in developed countries and 13% in developing countries are overweight or obese [4]. Therefore, childhood obesity is a global problem. Currently, 1 in 3 children in the United States is afflicted with overweight or obesity [5]. The prevalence of this epidemic exceeds 40% of children in southern European countries, with Bulgaria, Greece, Italy and Spain being the most affected countries [6,7,8,9]. The prevalence of this disease has increased in Spain three to four times in the last 20 years, causing a strong impact on chronic diseases, on health costs and on the quality of life of the population [10]. The Aladdin 2019 study, which was carried out in Spain recently, found a prevalence of overweight and obesity of 40.6% in the Spanish child population aged 6 to 9 years, of which 23.3% were overweight and 17.3% obese. Within obesity, 4.2% had severe obesity, and in turn, the percentage of obesity was higher in boys than in girls [11]. Within Spain, the province of Cádiz (southwest) has the highest prevalence of overweight and obesity (35% children aged 6 to 12 years) compared to the rest of Spain [12].

Obesity in early life is associated with a higher prevalence of various disease risk factors, such as prediabetes, increased blood pressure, accumulation of cholesterol in the arteries, increased oxidative vulnerability and sleep disorders [13,14,15,16,17]. Furthermore, paediatric and adolescent obesity is associated with obesity in adults [18], which increases the risk of multiples diseases in adulthood, such as diabetes, high blood pressure, dyslipidaemia, cardiovascular diseases, some types of cancer, musculoskeletal disease, obstructive sleep apnoea and fatty liver disease [19,20,21,22]. Life expectancy for 40-year-old people who suffer obesity is seven years less than that of one without excess weight [23].

On the other hand, lifestyle behaviours play a critical role in the development of childhood obesity [24,25]. The Jimenez-Pavón et al. study [26] analysed the relationship between dietary patterns and obesity and found that a dietary pattern that combined foods rich in fats, sugars and salt was directly related to an increased risk of obesity. Other studies [27,28,29,30,31] reported an inverse association between physical fitness and body weight status in children. Those children who presented a greater physical fitness in turn had a healthier body weight status. The same occurs with screen time; numerous studies [32,33] show that spending more than 2 h a day with a computer, mobile phone, tablet, playing video games or watching television considerably increases the probability of suffering from obesity in childhood. Likewise, sleep is also related to childhood overweight and obesity. According to a recent systematic review [34], children who slept less than 9 h a day had a greater risk of suffering from obesity than those who slept for more time. Therefore, carrying out healthy lifestyles (balanced diet, complying with the global recommendations for physical activity, reducing sedentary lifestyle and sleeping enough) keeps children healthy and reduces the probability of becoming overweight or obese [26,27,28,29,30,31,32,33,34,35].

In addition to lifestyles, physical fitness is well reported as a powerful marker of overall health and particularly related with overweight and obesity in children [36,37]. The role of fitness in children’s health is so relevant that even self-reported physical fitness, measured through the International Fitness Scale (IFIS), has shown to be a good tool to assess fitness and its relationships with health in children [36,38]. A systematic review found evidence to suggest that there are positive associations among physical activity, fitness, cognition and academic achievement in children [39]. On the other hand, a study carried out with Lithuanian children concluded by saying that obese and overweight children were less physically active and had lower physical fitness than normal-weight children, which shows that physical fitness is related to overweight and obesity [40]. Another systematic review found strong evidence for an inverse association between muscular fitness and total and central adiposity, and cardiovascular disease and metabolic risk factors in children and adolescents. In turn, they also found strong evidence for a positive association between muscular fitness and bone health and self-esteem [41]. The novelty of this study is the analysis of the combined role of lifestyle behaviours and self-reported physical fitness to check if there is association on body weight status in children. This is the first study in Spain that analyses the influence of these two dimensions on body weight status in children.

The hypothesis of the study is that the healthier the children’s lifestyle behaviours and self-reported physical fitness, the better their body weight status. Therefore, the objective of this article is to analyse the independent and combined association of lifestyle behaviours and self-reported physical fitness with body weight status in schoolchildren.

## 2. Materials and Methods

### 2.1. Study Design and Participants

The current study performed descriptive cross-sectional analysis of lifestyle behaviours and self-reported physical fitness of 864 schoolchildren between 8 and 9 years old enrolled in the third grade of primary education from 26 schools distributed throughout the province of Cádiz (southwest of Spain). An analysis to check if there was an independent and combined association between lifestyle behaviours and self-reported physical fitness on body weight status of the schoolchildren was carried out. The research implementation period was 2 months, which were the months of September and October 2018.

This target population (children from 8 to 9 years old, which is the age at which they attend the third year of primary education in Spain) was selected due to the need to detect and solve the problem of childhood obesity in early stages to promote healthier lifestyle habits, and thus achieve health benefits that allow correct weight management from childhood until adulthood, and because, at this age, children are very receptive to receiving information that implies changes in habits [42].

In order to select the schools, their management teams were informed by the research team about the research project. Of all the schools that were informed of the project, some preferred not to participate, but finally 26 schools spread throughout the province of Cádiz decided to participate and they were the ones that were recruited. To select the participating schoolchildren within each school, an informed consent was sent to the parents, who decided whether their children would participate or not. At the same time, children who did not want to participate even though their parents had signed the informed consent, were not forced to participate. The invitation to participate in the study was sent to 936 parents. Of these, among the parents who signed the informed consent and among the children who decided to participate, a total of 864 children were recruited. The response rate was 92.3%.

The inclusion criteria were as follows: children belonging to the selected school, who were in the third grade of primary school, regardless of their age, who had the informed consent signed by their parents or legal guardians, who responded to the questionnaires and who submitted to the anthropometric measurements. The exclusion criteria were that some of these requirements were not met.

The project was conducted in accordance with the Declaration of Helsinki and in accordance with Organic Law 3/2018, of December 5, on the Protection of Personal Data, Regulation (EU) 2016/679 of the European Parliament and of the Council, of 27 April 2016 (General Data Protection Regulation). Additionally, it was approved by the Andalusian Biomedical Research Ethics Committee, along with approval from the Delegation of Education and Science of the province of Cádiz (PI-007-2017, 21 March 2018). Parents or legal guardians were provided with a project information sheet and asked to sign the written informed consent. Participation was voluntary and schoolchildren could leave the study at any time. The methodological details have been previously published [42].

### 2.2. Calculation of the Sample Size

Although this article follows a cross-sectional design, the original sample size calculation was calculated to determine the effectiveness of an intervention on the prevalence of overweight and obesity in the study population, which is described in the study protocol already published [42].

### 2.3. Procedures and Assessments

#### 2.3.1. Anthropometric Measurements

Measurements were taken by trained research team members following the standardized International Society for the Advancement of Kinanthropometry (ISAK) procedure [43]. Body weight was measured with a mechanical scale sensitive to 100 g (SECA Colorata 760, Hamburg, Germany). Height was measured using a portable stadiometer with a precision of 0.1 cm (SECA 213, Hamburg, Germany). Body Mass Index (BMI) was calculated as weight in kilograms divided by height in meters squared and was used to determine the level of overweight or obesity. If the BMI of the students was between 18.44 and 19.84 according to Cole’s cut-off points [44], they were considered to be overweight, and if it was between 21.60 and 24.00, they were considered obese. In addition to this formula, the BMI categories were also calculated by estimating the percentiles according to their age based on the WHO recommendations, establishing that schoolchildren who were between the 50th and 80th percentile suffered overweight, and above the 85th percentile suffered obesity.

#### 2.3.2. Self-Reported Physical Fitness

The self-reported physical fitness of children and parents was measured using the IFIS scale (the International FItness Scale). The IFIS scale is a tool/questionnaire that has been translated into nine different languages, and which aims to assess the physical fitness of people in a self-reported/subjective way, without the need to carry out a battery of tests that are established to measure physical fitness objectively. This scale evaluates both overall physical condition as well as each one of its primary components specifically: cardiorespiratory fitness, muscular strength, speed-agility and flexibility. It has been previously validated with both children and adults [36,38], so its reliability is scientifically proven. There is a different version for adults and children, but both versions measure the same.

#### 2.3.3. Physical Activity and Screen Time

Physical activity and screen time were recorded through two questionnaires administered to schoolchildren and their parents, in order to have the information of the schoolchildren about themselves and that of the parents about their children. To record the levels of physical activity, both the students and their parents were asked about aspects such as travel to and from school, if they practiced extracurricular sports activities with a coach or monitor at that time, what activities, how often and for how long, if they practiced sports activities with family or friends at that time, how often and for how long, among others. Additionally, to record the screen time, both the schoolchildren and their parents were asked about the average daily time spent on television (TV), computer, mobile, tablet and video games during the weekdays and on weekends. The questionnaires were filled out at the same time they were administered, remembering what they usually did. The questionnaires administered were “Physical Activity Questionnaire: How do we move and feel?” for children and “Families: How do we eat and move?” for parents, which were the ones previously used by the Previene Study in Granada and POIBA Study in Barcelona [45,46].

#### 2.3.4. Eating Habits

Eating habits were recorded through two questionnaires, one applied to schoolchildren, who answered about themselves, and another applied to parents, who answered about their children. The administered questionnaires were “Food Consumption Frequency Questionnaire: How do we eat?” for children and “Families: How do we eat and move?” for parents, which were the ones previously used in the Previene Study in Granada and POIBA Study in Barcelona [45,46]. The frequency of food consumption was recorded (average daily/weekly consumption of processed baked goods, fried foods, snacks, sugary soft drinks, fruits, vegetables, packaged juices, sandwiches, dairy products, carbohydrates, meat, cold cuts, fish, and vegetables, among others), the average frequency with which schoolchildren ate breakfast before going to school and the average frequency with which they ate outside the home (in fast food establishments or restaurants). Like the previous ones, these questionnaires were also completed at the same time they were administered, using memory and remembering what they used to do regarding their eating habits.

#### 2.3.5. Hours of Sleep

Schoolchildren’s daily hours of sleep, both for Monday to Friday as well as for weekends, were also recorded through a questionnaire administered to parents called “Families: How do we eat and move?”, which was adapted from the one previously used in a similar sample in the Previene Study in Granada and POIBA Study in Barcelona [45,46]. Parents registered the average time their children normally went to bed and got up on weekdays and weekends. To assess compliance with the recommendation for healthy habits, the cut-off points recommended by the Canadian 24-Hour Movement Guidelines [47] were established. This questionnaire was also filled out at the same time it was applied, using memory and remembering the average time at which their children used to go to bed at night and got up in the morning, both during the week and at the weekends.

#### 2.3.6. Lifestyle Behaviours

Lifestyle behaviours were evaluated from the variables that influenced the lifestyle of schoolchildren, such as the variables that influenced the levels of physical activity of schoolchildren (going and coming home from school, hours of Physical Education at school, attendance at extracurricular sports activities, practice of sports activities with family or friends, among others), the variables that influenced the eating habits of schoolchildren (frequency of food consumption, frequency of breakfast, frequency of meals in fast food establishments or restaurants, among others), the variables that influenced the levels of sedentary lifestyle of schoolchildren (screen time) and those that influenced sleeping habits (hours of sleep).

### 2.4. Statistical Analysis

A descriptive analysis of the outcome variables was performed with measures of centrality, dispersion and 95% confidence intervals (CI), and a frequency analysis to obtain percentages and prevalence of the categorical variables. The data are presented as mean ± standard deviation, unless otherwise stated. The outcome variables were log-transformed to get a normal distribution. To calculate the association between self-reported physical fitness, having breakfast, screen time and sleep time on body weight status, an ANOVA test was realized. In order to verify the influence of lifestyle behaviours and self-reported physical fitness on the body weight status of schoolchildren, a combined score of lifestyle behaviours and self-reported physical fitness was calculated. To calculate the combined score, a variable was taken from each category (one from eating habits, which was breakfast frequency, one from sedentary behaviours, which was hours of TV on weekends, one from sleep time, which was hours of sleep on weekends, and one of self-reported physical fitness, which was the general physical condition), the dichotomous variable of each of them was calculated, and a sum of the four dichotomous variables was made, giving rise to a result variable with 5 categories. This result variable ranged from 0 to 4, and was re-categorized into another variable with 3 categories, which were called “healthy, moderately healthy and unhealthy”. Once this variable was calculated, it was checked whether there was an association between this variable (combined score) and body weight status. For this, an ANOVA test was performed. Logistic regression was then performed to analyse the influence of the combined score on body weight status. A logistic regression was realized to analyse the influence of the combined score on body weight status. The statistical analyses were performed using IBM SPSS Statistics (version 24.0) and the level of significance was set to 0.05.

## 3. Results

### 3.1. Baseline Characteristics

Of the 864 participants, as can be seen in Table 1, 47.7% were girls, 35.2% were overweight (12.9% obesity and 3.9% severe obesity), 25% had a BMI ≥ 97th percentile for age and sex, and the mean age of the participants was 8.42 ± 0.34 years.

### 3.2. Independent Associations of Lifestyle Behaviours

A positive association was found between lifestyle behaviours of schoolchildren (levels of physical activity, eating habits, levels of sedentary lifestyle and sleeping habits) and their body weight status. The healthier the schoolchildren’s lifestyle behaviours, the better their body weight status.

Regarding breakfast frequency, a positive association was found between this variable and body weight status of schoolchildren, both for the total sample and segmented by sex. As can be seen in Figure 1, schoolchildren who had breakfast more regularly had a better body weight status than those who skip breakfast or had breakfast less frequently. The schoolchildren who had breakfast always or almost always (6 or 7 days a week)/many times (3 to 5 times a week)/sometimes (less than 3 times a week)/never or almost never (once a week or less), had a mean BMI of 18, 19, 19.5 and 20.6, respectively, F = 7.5 (3) *p* < 0.001.

There was also an association between screen time (parents reported) and body weight status. As shown in Figure 2, schoolchildren who spent less than 2 h/between 2 and 4 h/more than 4 h a day in front of TV on weekends, had a mean BMI of 17.8, 18.7 and 19.3, respectively, F = 6.7 (2) *p* = 0.001. When separated by sex, in the case of boys, the differences were statistically significant F = 7.1 (2) *p* = 0.001. Boys who spent less than 2 h/between 2 and 4 h/more than 4 h a day in front of TV on weekends, had a mean BMI of 17.8, 18.9 and 20.5, respectively. However, the results were not statistically significant for the girls. Those who spent less than 2 h/between 2 and 4 h/more than 4 h a day in front of TV on weekends, had a mean BMI of 17.8, 18.4 and 18.2, respectively, F = 1.2 (2) *p* = 0.296.

Finally, a positive association was also found between sleep time and body weight status. As can be seen in Figure 3, schoolchildren who slept less than 9 h/between 9 and 11/more than 11 h a day on weekends, had a mean BMI of 20.4, 18.2 and 18.1, respectively, F = 3.2 (2) *p* = 0.041. However, although the results followed the same trend, no significant results were found when segmented by sex. Boys who slept less than 9 h/between 9 and 11/more than 11 h a day on weekends, had a mean BMI of 21, 18.3 and 17.7, respectively, F = 2.9 (2) *p* = 0.054. Additionally, girls who slept less than 9 h/between 9 and 11/more than 11 h a day on weekends, had a mean BMI of 19.8, 18 and 18.3, respectively, F = 1.2 (2) *p* = 0.300.

### 3.3. Independent Associations of Self-Reported Physical Fitness

There was a positive association between several of the self-reported physical fitness and body weight status. As shown in Figure 4, the higher the self-reported physical fitness, the better the body weight status of schoolchildren. In the case of general physical fitness, it was observed that the schoolchildren who answered that they had a very good/good/acceptable/bad general physical fitness had a mean BMI of 17.5, 18.04, 19.6 and 21.2, respectively, F = 15.2 (3) *p* < 0.001. The answers regarding cardiorespiratory fitness and muscular strength followed the same trend, although the differences were not significant: F = 1.5 (3) *p* = 0.21 and F = 0.6 (3) *p* = 0.61, respectively. However, the difference in the mean BMI across the answer categories of speed-agility were significant: F = 22.5 (3) *p* < 0.001 ranging from 17.3 to 20.8. The schoolchildren who answered that they had a very good/good/acceptable/bad speed-agility had a mean BMI of 17.3, 18.5, 19.4 and 20.8, respectively. Regarding flexibility, significant differences were also found with respect to BMI. The schoolchildren who answered that they had a very good/good/acceptable/bad flexibility had a mean BMI of 17.4, 18.3, 18.7 and 18.8, respectively, F = 8.2 (3) *p* < 0.001. The same happened when it was segmented by sex (general physical fitness, F = 10.7 (3) *p* < 0.001; cardiorespiratory fitness, F = 1.3 (3) *p* = 0.26; muscular strength, F = 0.4 (3) *p* = 0.71; speed-agility, F = 14.7 (3) *p* < 0.001; flexibility, F = 2.4 (3) *p* = 0.07 for boys, and general physical fitness, F = 5.03 (3) *p* = 0.002; cardiorespiratory fitness, F = 0.5 (3) *p* = 0.69; muscular strength, F = 1.04 (3) *p* = 0.37; speed-agility, F = 8.8 (3) *p* < 0.001; flexibility, F = 8.1 (3) *p* < 0.001 for girls).

### 3.4. Combined Associations of Lifestyle Behaviours and Self-Reported Physical Fitness

Once these results were obtained, in order to verify the influence of lifestyle behaviours and physical fitness on the body weight status of schoolchildren, a combined score of lifestyle behaviours and self-reported physical fitness was calculated. For this, a variable was selected from each category (eating habits, sedentary behaviour, sleep time and self-reported physical fitness). Then, a summation of variables was performed, and finally, the outcome variable was re-categorized into three categories according to the lifestyle behaviours that the schoolchildren follow (good/regular/bad lifestyle behaviours). This means that the schoolchildren who had a good/regular/bad score, had healthier/regular health/less healthy lifestyle behaviours and good/regular/bad self-reported physical fitness.

Effectively, as shown in Figure 5, schoolchildren who followed healthier lifestyle behaviours and presented good values of physical fitness had a better body weight status. Schoolchildren who presented a good/regular/bad score in the combined score of lifestyle behaviours and physical fitness had a mean BMI of 17.4, 18.8 and 21.7, respectively, F = 18.1 (2) *p* < 0.001. The same happened when it was segmented by sex. Boys who presented a good/regular/bad score in the combined score of lifestyle behaviours and physical fitness had a mean BMI of 17.2, 19.2 and 21.1, respectively, F = 14.5 (2) *p* < 0.001. Additionally, girls who presented a good/regular/bad score in the combined score of lifestyle behaviours and physical fitness had a mean BMI of 17.6, 18.3 and 22.4, respectively, F = 6.3 (2) *p* = 0.002.

After realising the logistic regression, it was observed that the schoolchildren who had a regular lifestyle behaviours and regular physical fitness (a regular assessment on the combined score) had a 2.015 times higher risk of being overweight or obese than those who followed healthy lifestyle behaviours and had a good physical fitness (had a good assessment on the combined score) (*p* < 0.001). Additionally, schoolchildren who had bad lifestyle behaviours and bad physical fitness (a bad assessment on the combined score) had a 10.34 times higher risk of being overweight or obese than those who followed healthy lifestyle behaviours and had a good physical fitness (had a good assessment on the combined score) (*p* = 0.004).

These results demonstrate the importance of following healthy lifestyle behaviours and having a good physical fitness to maintain a correct and healthy body weight status at school age.

## 4. Discussion

This study aimed to verify the association between lifestyle behaviours and self-reported physical fitness (individually and combined) with body weight status in schoolchildren from the province of Cádiz. Particularly, it was found that there were independent and combined associations of lifestyle behaviours and physical fitness with body weight status in schoolchildren.

Lifestyle behaviours are key determinants in the developing of healthy/unhealthy body weight status. It has been shown in this study that self-reported physical fitness predicted overweight and/or obesity in schoolchildren, as well as the combination of lifestyle factors in this cohort, and can thus be a valuable screening tool. In addition, the combined score of lifestyle behaviours and self-reported physical fitness as a tool may also be valuable for predicting childhood overweight and obesity.

One of these lifestyle behaviours is related with nutrition habits such as breakfast, which is one of the more controversial meals, considered a key meal of the day on some occasions and questioned in others. Most of the studies associated breakfast with better body weight control and healthy cardiometabolic risk indicators in children [48,49,50]. However, other studies suggest inverse or controversial relationships [51,52,53]. Several studies have shown a direct relationship between breakfast skipping and being overweight or obese [49,54]. According to López-Sobaler et al. [48], the breakfast of the Spanish population can be improved, since a high percentage skip breakfast, have an insufficient breakfast or incorporate inadequate foods. Our findings concur with those reported by López-Sobaler et al., Szajewska et al. and Monzani et al. [48,49,50,54], who suggest breakfast is associated with better body weight control and skipping breakfast is directly related with being overweight or obese.

Screen-based sedentary behaviours (watching TV, playing video games, using mobile phones, tablets and computers) are common among young people, most children failing to meet guidelines of <2 h of television per day [55]. This is of concern given the positive associations between increased levels of screen time, sedentary behaviour and adverse health outcomes [56]. The results found in the present study are in line with this; schoolchildren who spent more than 4 h a day on screen time had a worse body weight status than those who spent between 2 and 4 h or less than 2 h. Therefore, as supported by previous studies, a negative association was found in the present study between screen time and the body weight status of the schoolchildren.

Insufficient sleep in childhood has been shown to significantly raise the risk of overweight and obesity [13,57]. These observations suggest that insufficient sleep affects food intake via hedonic rather than homeostatic processes [58]. Evidence to date points to food intake rather than activity as the primary pathway [59]. Epidemiological studies in children have identified an inverse relationship between sleep duration and energy intake [60,61]. According to these studies, a positive association was found between hours of sleep and body weight status. Schoolchildren who slept more hours a day had a better body weight status than those who slept fewer hours.

On the other hand, according to several studies [62,63,64], perceived physical fitness is considered a powerful marker of health already in youth. This supports the results obtained in the present study, where a positive association was observed between physical fitness and body weight status. It was found that the higher the physical fitness, the better the children’s body weight status. Along the same lines are the results of the cross-sectional study carried out by Ruiz-Montero, J.P. et al. [65], where a positive association between self-reported physical fitness and body weight status was found in a group of adolescent secondary school students. According to this study, a higher body mass index in adolescents was associated with a worse general physical fitness, cardiorespiratory fitness, muscle strength, speed-agility and flexibility.

A recent systematic review [35] showed that lifestyle behaviour interventions were generally effective at reducing excess weight in children and adolescents. Absolute reductions in BMI z-score of 0.2 or more were observed in the intervention groups, compared with virtually no reduction in the control groups. In turn, a community intervention focused on improving the lifestyle behaviours of schoolchildren between 10 and 12 years old, carried out in Australia, showed a 6% reduction in the prevalence of overweight and obesity in a follow-up at 3 years after the intervention among schoolchildren in the intervention group [66]. Furthermore, in a study carried out by Anna-Kaisa, K. et al. [67] a positive association was found between physical fitness and body weight status in 8-year-old children. Children who were overweight or obese had an impaired performance in tests requiring muscle endurance, balance, explosive power of lower extremities, upper body strength, endurance and speed and agility in both genders and exercise capacity in boys. These findings support the positive association found in the present study between the combined score created for lifestyle behaviours and physical fitness on body weight status of schoolchildren. Children who followed healthier lifestyle behaviours and had better physical fitness had better body weight status than those who followed unhealthy lifestyle behaviours and presented bad physical fitness.

As a possible limitation of the study, it should be noted that the physical fitness of schoolchildren and family members was not objectively assessed but doing so subjectively using the self-report questionnaire of perceived physical condition IFIS. It was decided to evaluate the physical fitness in this way due to the complexity of bringing together schoolchildren, and especially their families to perform a series of physical tests. As a positive aspect to support this decision, point out that the IFIS has been previously validated and used successfully in this population group [38]. Another possible limitation of this study is that it is a cross-sectional study, where a single measurement is collected at a specific point in time, so it cannot be compared with a previous or subsequent measurement, but since what has been done is to check if there is association between the study variables and the body weight status of the schoolchildren by way of description, there should be no problem when using this design. In future studies, the pre and post intervention measurements will be compared to see if there have been changes with the performance of the intervention or not, and to verify its efficacy or not.

Among the strengths of this project, it should be noted that there is a large and representative sample of boys and girls aged 8 to 9 from all over the province of Cádiz, with schools spread throughout the province. In addition, it had the response of the schoolchildren and their families, which made it possible to counteract the information of the children and parents, obtaining more complete and truthful information from the study sample.

## 5. Conclusions

In accordance with the objective proposed in the present study, it has been shown that there is an independent and combined association between lifestyle behaviours and physical fitness on the body weight status of schoolchildren in the third grade of primary education in the province of Cádiz.

The better the lifestyle behaviours and the physical fitness that schoolchildren lead, the better their body weight status. Therefore, it is suggested that strategies or interventions be carried out in schools, involving teachers, schoolchildren and their families, to get children to adopt healthy lifestyles and have a good physical fitness from an early age. This will allow them to maintain a healthy body weight status both in childhood and in adulthood, preventing overweight and obesity, as well as various health problems associated with this disease.

## Figures and Tables

**Figure 1 nutrients-14-01208-f001:**
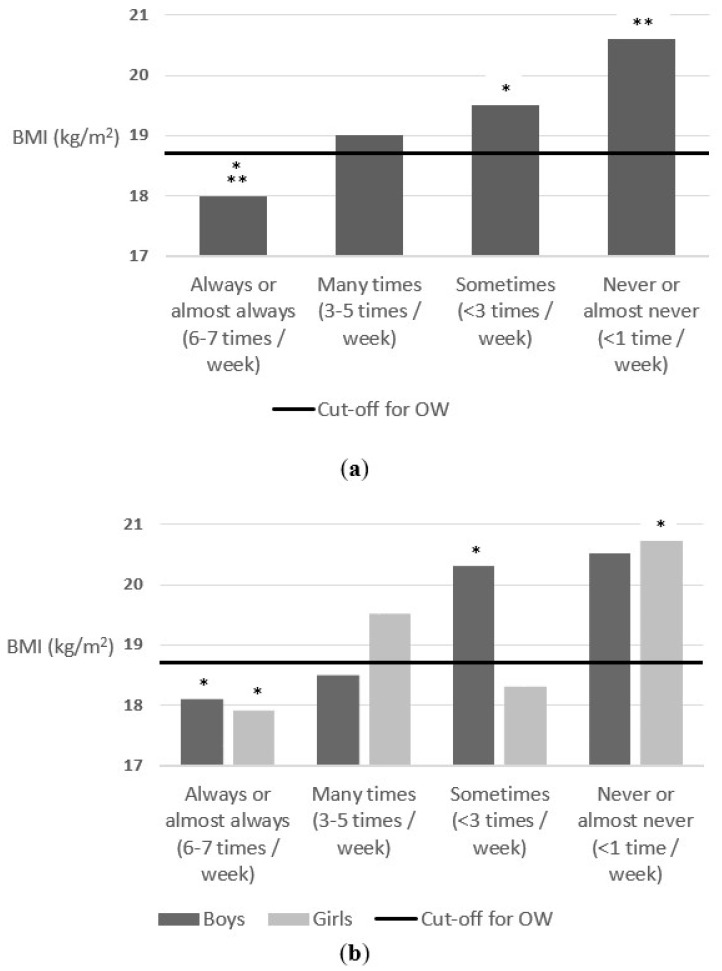
(**a**) Association between breakfast frequency and body weight status (parents reported); (**b**) Association between breakfast frequency and body weight status (parents reported) segmenting by sex. The cut-off for OW line indicates the beginning of the OW category according to Cole’s cut-off points [44]. * *p*-value < 0.05 and ** *p*-value < 0.01.

**Figure 2 nutrients-14-01208-f002:**
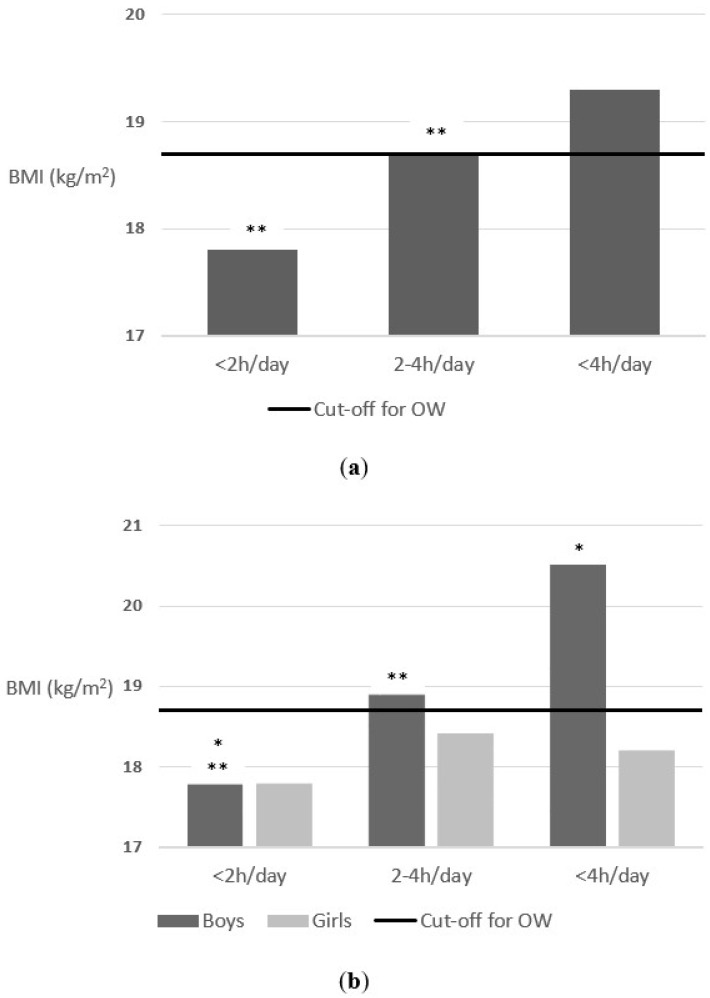
(**a**) Association between time in front of TV on weekends a day and body weight status (parents reported); (**b**) Association between time in front of TV on weekends a day and body weight status (parents reported) segmenting by sex. The cut-off for OW line indicates the beginning of the OW category according to Cole’s cut-off points [44]. * *p*-value < 0.05 and ** *p*-value < 0.01.

**Figure 3 nutrients-14-01208-f003:**
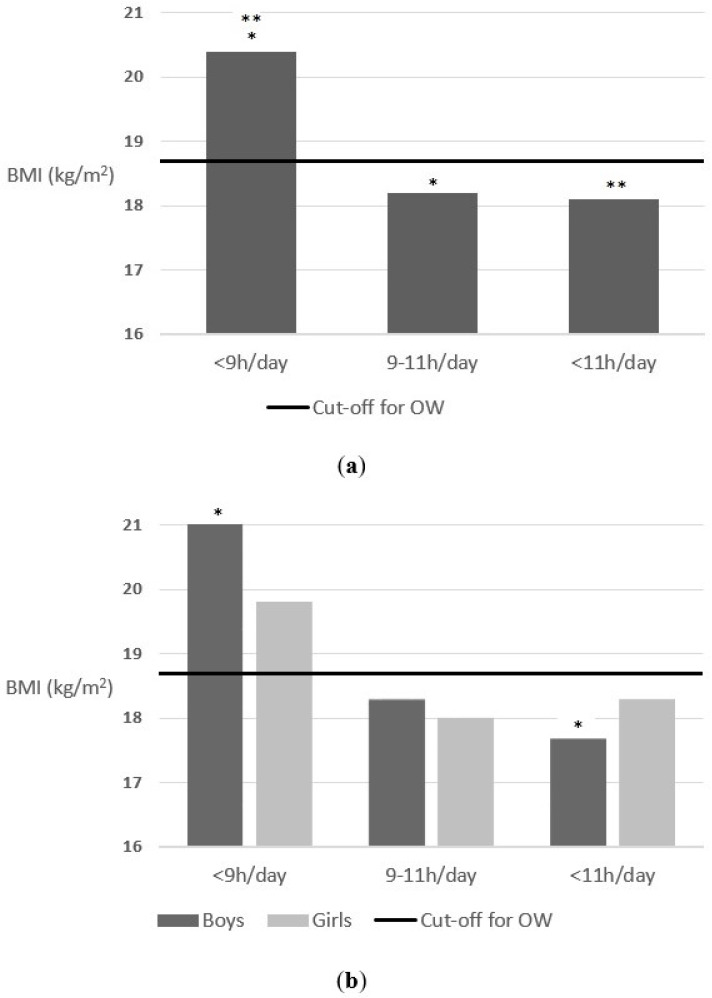
(**a**) Association between hours of sleep a day on weekends and body weight status (parents reported); (**b**) Association between hours of sleep a day on weekends and body weight status (parents reported) segmenting by sex. The cut-off for OW line indicates the beginning of the OW category according to Cole’s cut-off points [44]. * *p*-value < 0.05 and ** *p*-value < 0.05.

**Figure 4 nutrients-14-01208-f004:**
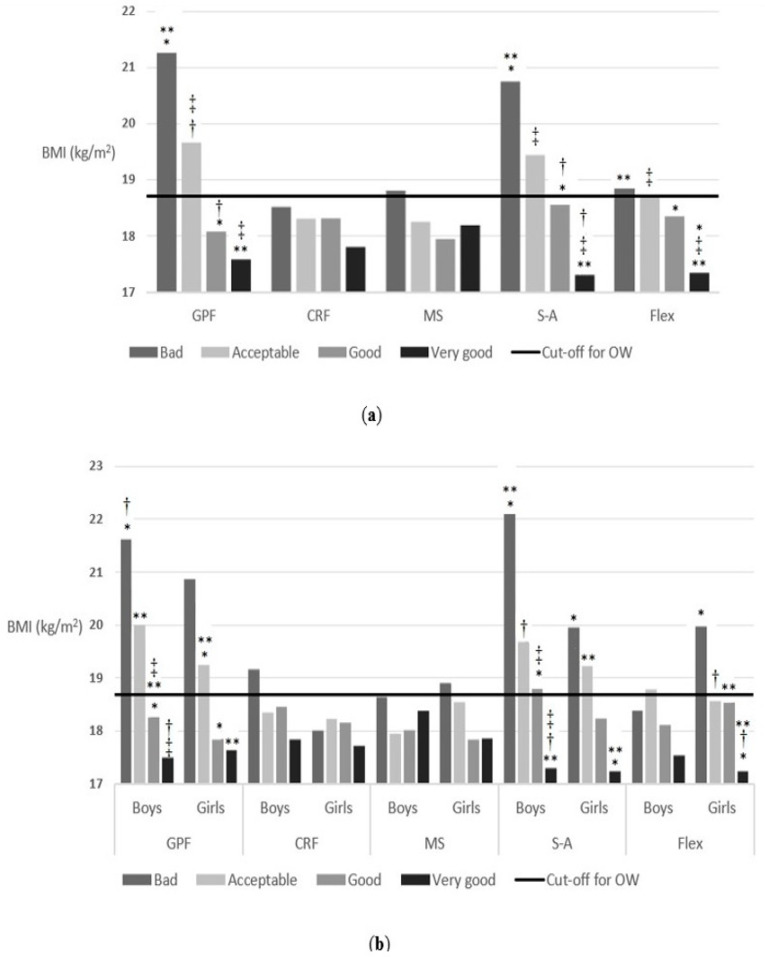
(**a**) Association between self-reported physical fitness and body weight status (children reported); (**b**) Association between self-reported physical fitness and body weight status (children reported) segmenting by sex. The cut-off for OW line indicates the beginning of the OW category according to Cole’s cut-off points [44]. * *p*-value < 0.05, ** *p*-value < 0.05, ^†^ *p*-value < 0.05 and ^‡^ *p*-value < 0.05. Abbreviations: CRF, cardiorespiratory fitness; Flex, flexibility; GPF, general physical fitness; MS, muscular strength; S-A, speed-agility.

**Figure 5 nutrients-14-01208-f005:**
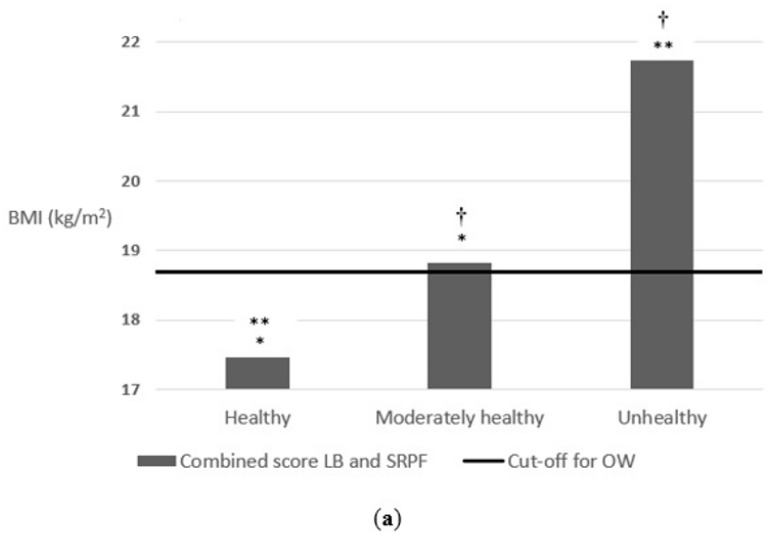
(**a**) Association between the combined score of lifestyle behaviours and self-reported physical fitness on body weight status; (**b**) Association between the combined score of lifestyle behaviours and self-reported physical fitness on body weight status segmenting by sex. The cut-off for OW line indicates the beginning of the OW category according to Cole’s cut-off points [44]. * *p*-value < 0.001, ** *p*-value < 0.001, ^‡^ *p*-value < 0.01 and ^†^ *p*-value < 0.05. Abbreviations: LB, lifestyle behaviours; SRPF, self-reported physical fitness.

**Table 1 nutrients-14-01208-t001:** Descriptive characteristics of schoolchildren by sex.

Variable	Total Sample *n* = 864	Male *n* = 452	Female *n* = 412
**Physical characteristics**	**Mean (SD) or Percentage**		
Age (years)	8.42 ± 0.34	8.43 ± 0.35	8.42 ± 0.33
Weight (kg)	31.33 ± 7.62	31.92 ± 8.13	30.77 ± 7.07
Height (cm)	130.68 ± 5.99	131.53 ± 6.12	129.88 ± 5.75 **
BMI (kg/m^2^)	18.17 ± 3.44	18.27 ± 3.55	18.08 ± 3.44
BMI Status (%) (UW/NW/OW/Ob)	5/60/22/13	4/62/20/14	7/56/24/13
**Self-reported physical fitness**			
General physical fitness (%) (B/A/G/VG)	2/15/37/46	2/16/33/49	2/14/41/43
Cardiorespiratory fitness (%) (B/A/G/VG)	7/21/33/39	6/20/30/44	7/22/37/34 *
Muscular strength (%) (B/A/G/VG)	3/19/33/45	4/20/25/51	3/17/41/39 ^†^
Speed/Agility (%) (B/A/G/VG)	4/15/30/51	3/12/31/54	5/18/29/48
Flexibility (%) (B/A/G/VG)	14/24/25/37	20/28/25/27	7/20/25/48 ^†^
**Feeding (children reported)**			
Breakfast (%) (Yes/No)	91/9	89/11	92/8
To eat in a restaurant (%) (Al/MT/S/N)	2/28/64/6	3/30/61/6	2/25/67/6
**Feeding (parents reported)**			
Breakfast (%) (Al/MT/S/N)	86/6/5/3	87/6/5/2	85/6/6/3
To eat in a restaurant (%) (N/S/MT/Al)	30/67/2/1	29/69/2/0	30/67/2/1
**Screen time (children reported)**			
TV and VG weekdays (%) (<2 h/2–4 h/>4 h)	61/23/16	55/25/20	67/21/12 **
TV and VG weekends (%) (<2 h/2–4 h/>4 h)	46/28/26	38/30/32	54/26/20 ^†^
PC and MP weekdays (%) (<2 h/2–4 h/>4 h)	72/17/11	68/18/14	76/17/7 **
PC and MP weekends (%) (<2 h/2–4 h/>4 h)	60/21/19	54/23/23	67/19/14 ^†^
**Screen time (parents reported)**			
TV weekdays (%) (<2 h/2–4 h/>4 h)	89/10/1	87/11/2	91/8/1
TV weekends (%) (<2 h/2–4 h/>4 h)	64/33/3	63/34/3	64/33/3
VG weekdays (%) (<2 h/2–4 h/>4 h)	97/2/1	95/4/1	98/1/1 **
VG weekends (%) (<2 h/2–4 h/>4 h)	85/13/2	78/17/5	90/8/2 ^†^
PC and MP weekdays (%) (<2 h/2–4 h/>4 h)	97/2/1	98/1/1	96/3/1
PC and MP weekends (%) (<2 h/2–4 h/>4 h)	86/12/2	85/13/2	87/11/2
**Hours of sleep**			
Weekdays (h/day)	10.00 ± 0.55	10.05 ± 0.52	9.96 ± 0.58 *
Weekends (h/day)	10.38 ± 0.89	10.33 ± 0.95	10.43 ± 0.83
Weekdays (%) (<9 h/9–12 h/>12 h)	2/98/0	1/99/0	2/98/0
Weekends (%) (<9 h/9–12 h/>12 h)	2/96/2	2/96/2	2/96/2

Values are presented as mean ± standard deviation or percentages. *t*-test and γChi square statistics was applied. Abbreviations: A, acceptable; Al, always; B, bad; BMI, body mass index; Cm, centimetres; G, good; h/day, hours per day; Kg, kilograms; Kg/m^2^, kilogram per square meter; MP, mobile phone; MT, many times; N, never; NW, normal weight; Ob, obesity; OW, overweight; PC, personal computer; S, sometimes; SD, standard deviation; UN, underweight; VG, video games; VG, very good; (<2 h, less than two hours; 2–4 h, between two and four hours; >4 h, more than four hours; <9 h, less than nine hours; 9–12 h, between nine and twelve hours; >12 h, more than twelve hours. * *p*-value < 0.05, ** *p*-value < 0.01 and ^†^ *p*-value < 0.001.

## Data Availability

The data supporting the current findings can be found upon request to the corresponding author in the repository of scientific data from the local university.

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
