# Peer review of "Independent and Combined Association of Lifestyle Behaviours and Physical Fitness with Body Weight Status in Schoolchildren"

_nutrients, 2022, doi:10.3390/nu14061208_

Round 1

Reviewer 1 Report

The authors present a descriptive cross-sectional study analyzing association of lifestyle behaviors and self-reported physical fitness on body weight in 3rd graders from Cadiz. They show striking association of self-reported physical fitness with weight status, as well as independent and additive effect of other lifestyle behaviors (breakfast eating, screen time, sleep). The results are interesting and support the use of these assessments to screen for at-risk children that could be helped by interventions aimed at improving physical fitness & lifestyle behaviors. However, the presentation could use some editing and clarification, as detailed below.

Abstract

Line 30-31 No evidence has been reported about the combined role of lifestyles and physical fitness on body weight.  I think this is overstated. Many studies have shown exercise & diet effects. What is novel in this study is the finding that self-reported physical fitness is representative of healthy lifestyle and thus is associated with lower incidence of overweight/obesity.

Lifestyles + physical fitness---does “lifestyles” mean diet? Define lifestyles. Also clarify that physical fitness is self-reported.

Line 36-37 To obtain the body weight status, the children were measured by weight and 36 height. “body weight status”---did they use BMI? Then say BMI

Line 38 “Bad”. Please replace with a “less healthful” or define certain parameters of the lifestyle

You do not mention in the abstract that a composite score was used---this should be clarified.

Methods:

Study design (109-115). This is confusing/misleading. You mention first that this was an experimental design and then later that this was descriptive data from before the intervention. Start with saying this was descriptive cross sectional, and then can add that this cohort was also part of the experimental design.

117-118. What does this mean, that the parents were part of the same study subject?

Sample size calculation---you can summarize in a sentence or two or refer to publication of the experimental study (if published) since the sample size is not relevant to this analysis

2.3.1---nothing is reported in the results on the sociodemographic parameters other than parental education level---and this is not even discussed in results., so why is this in the methods? Were these used in the analyses? If not, should remove. Or add to “baseline characteristics” and discuss in more detail in results

175: “IFIS” more details need to be given on the physical fitness tool. I know this was previously reported, but this is the crux of the paper, and readers shouldn’t have to refer to another paper to get details.

For diet/sleep/activity/screen time, were these recorded on one day (using memory of kids/parents) or did they complete for a week or more? It is unclear how this was done.

210: which component of “feeding” was used?

211-212 How were the “good/regular/bad” categories determined? More info needed here. I also think different words (non-judgmental) should be used. This may deserve a figure/table but at the least, a good explanation

Results

Line 223-235. Suggestion for clarity---If the effect was same in boys/girls, not needed to present the full data and just state that it was seen in both boys/girls, and refer to the Figure

For screen time---was it child reported or parent reported that was used? Both are shown in the table and seem quite different. Please clarify

296, 301 I’m not sure what “punctuation” means here.

Discussion

Again, I think saying that the association of weight with the combination of lifestyle and physical fitness has never been reported is overstatement. I would argue that the strength here is that self-reported physical fitness---on its own---predicted OW/Ob as well as, or better than, the combination of lifestyle factors in this cohort, and can thus be a valuable screening tool. If it has not been reported, the composite score of lifestyle behaviors as a tool may also be novel

Line 386---physical fitness. We don’t actually know the child’s physical fitness, should be restated as “perceived physical fitness” as there were no objective measures

Limitation---while it was spread over Cadiz, we don’t know if this is representative of Spain as a whole, Europe, global. All the kids were the same age, so don’t know how this would work in younger/older kids.

Author Response

Dear reviewer,

Thank you very much for the opportunity to review this manuscript. The article has been rewritten with special attention to the suggestions of the reviewers. All the changes made have been highlighted in yellow. Please see the attachment.

Reviewer 2 Report

First question: that it is necessary to appear in the title ''Previene-Cádiz''?

Introduction - the authors mention that many schoolchildren are obese in schools in Europe, why was not specified the values ​​of obesity from other countries outside Europe ? We know very well that obesity scares the whole world once it has taken the form of an epidemic in the most developed countries: Hungary, Iceland, Chile, Canada, Australia, America, etc. I think it would be good to present in this chapter other countries outside Europe because it is a global problem.

Also, in this chapter, please highlight very clearly what was the novelty of this study.

What was the reason why only the third graders were chosen? Is there an explanation for their choice? Also, out of a total of 864 children, did everyone want to participate? were there no refusals from them or their parents? It is very important for the authors to specify very clearly the methodology for conducting the research, respectively for recruiting the subjects. From what is presented, it is not very clear. Also, what was the period of conducting the research ...

Self-Reported Physical Fitness
Please complete this assessment tool with additional information, there is very little information in this subchapter, especially since physical condition is a very important variable in this study. IFIS - appears in the introduction, with very little information, appears in this subchapter and is not developed, please add information necessary for a very clear understanding by readers.

''Physical activity and screen time were recorded by means of a self-report by the 180 schoolchildren and observation of the parents using a built-in questionnaire....'', Ok, but what's the name of the questionnaire? Even if it is not an important variable of the research, please fill in the necessary information.....also for sleep questionnaire

Diet

What were those 2 questionnaires? please provide details, do not present general data .... diet has a very important role in this research

Lifestyle- I did not understand how lifestyle was assessed, ........it is an important variable in this study. Was there no questionnaire or assessment tool? In the methods chapter I did not see anything presented about this variable, only in the subchapter "Statistical Analysis". Please enter in the methodology clear information about this variable...

Results - for each abbreviation please present under the table their full name

As I said before the authors have to present adjacent information about the life style, they present some results of this variable but we still don't know how it was evaluated, it is very difficult for the readers to figure out how it was evaluated.

It would have been interesting to see what is the level of fitness for the subjects involved in the research, the results show the correlations between the different parameters, but we do not know exactly what was the initial level of fitness for participants.

One worrying thing is that the authors stated on line 110 that they have an "intervention group and a control group." Where are the two groups? what was done with each group? We did not see in the article comparative analyzes between the groups and the content of the implemented programs, it is a big confusion here that the authors should detail.

Similarly, comparative information between the experimental group and the control group should be presented in terms of discussions, but ... this information does not exist.

In conclusion, I can say that the idea of ​​the article is not bad, but the authors should rewrite the article and follow exactly what is written in the methodology presented. The problem related to the experimental group and the control group must be detailed and presented with data, respectively with what exactly was intervened in both groups. If you want to focus on questionnaires, then there is no need for intervention. 

Author Response

(The authors gave the same response as above.)

Reviewer 3 Report

A nicely conducted and well presented study.  Some additional information would further strengthen it:

  • were the schools the total in the province, or a selection. If a selection, please give the selection criteria, and assess any bias resulting.
  • consent rate
  • whether non-participating children were different in key demographics
  • inherent bias of selection criterion of 'having consented' (which of course is necessary), but did this bias against any groups such as immigrants, lower educated families, fractured families.

Author Response

(The authors gave the same response as above.)

Round 2

Reviewer 1 Report

Authors have addressed all the comments and improved the manuscript. 

Reviewer 2 Report

The authors have substantially improved the article, now its essence is much clearer.

Regarding the novelty of the study, can you mention that it is the first study in Spain that analyzes the two variables, i.e lifestyle and self-reported physical fitness? Are no more similar or close studies on Spanish with this topic?

I understood they remained in the end 864 children, the question was how many were applied, or was the project presented in the initial phase? how many parents were asked in the initial phase? Also, please specify exactly the months of implementation, not just how many there were.

IFIS scale - does it show a score? if so, please submit it.

From this comment: ''One worrying thing is that the authors stated on line 110 that they have an "intervention group and a control group." Where are the two groups? what was done with each group? We did not see in the article comparative analyzes between the groups and the content of the implemented programs, it is a big confusion here that the authors should detail''......The authors' answer did not convince me, as I said earlier in this research, there is no experimental group, and no control group, it will probably be a mistake made by the authors?......

Thank you!

Author Response

Thank you very much for reviewing and considering our article again. We have rewritten it paying special attention to your suggestions. All changes made have been marked up using the “Track Changes” function of the MS Word. We hope that the article is now to your liking and is ready to be published in the journal. Please see the attachment. Thank you very much for your contributions.
